# Evaluating Two Brief Motivational Interventions for Excessive-Drinking University Students

**DOI:** 10.3390/bs14050381

**Published:** 2024-05-01

**Authors:** Lee M. Hogan, W. Miles Cox

**Affiliations:** 1North Wales Clinical Psychology Programme, Bangor University, Bangor LL57 2DG, UK; 2Substance Misuse Services, Betsi Cadwaladr University Health Board, Rhyl LL18 3EY, UK; 3School of Psychology and Sport Sciences, Bangor University, Bangor LL57 2DG, UK; m.cox@bangor.ac.uk

**Keywords:** alcohol consumption, computerized brief intervention, gender difference, motivation, university students

## Abstract

Objective: Two brief computerized motivational interventions for excessive-drinking university students were evaluated. Method: Participants (*N* = 88, females = 61.5%, mean age = 21.05 years) were randomly assigned to a control group or one of two experimental groups: Computerized Brief Intervention (CBI) or Computerized Brief Intervention-Enhanced (CBI-E). CBI followed the principles of Motivational Interviewing to motivate participants to change their drinking behavior. CBI-E additionally used the principles of Systematic Motivational Counseling to identify and discuss with participants their dysfunctional motivational patterns that were interfering with their attainment of emotional satisfaction. At baseline and a three-month follow-up, the participants completed a battery of measures of alcohol consumption and related problems. Results: At baseline, the participants were confirmed to be heavy drinkers with many drink-related negative consequences. Males and females responded differently to the interventions. During follow-up, males’ alcohol use was ordered: CBI-E < CBI < Controls. The females in all three groups reduced their alcohol use, but there were no significant group differences. Conclusions: Males responded to the interventions as expected. For females, the assessment itself seemed to serve as an effective intervention, and there were no post-intervention differences among the three groups. Suggestions for future research using CBI and CBI-E are discussed.

## 1. Introduction

University students’ excessive consumption of alcohol (drinking more than 14 units of alcohol per week), often in binges (“drinking heavily over a short space of time” or “drinking to get drunk”; National Health Service), is a major problem [1,2,3,4,5]. When students drink in a hazardous or harmful manner, this can lead to a variety of short-term and long-term academic, economic, medical, psychological, and social consequences [4,6]. Some excessive-drinking students will eventually resolve their problems with alcohol. Other students, however, will risk developing into dependent drinkers [7]. Clearly, therefore, interventions are needed to tackle the problem of university students’ drinking. Over the years, various programs have, in fact, been developed to help students control their drinking, but issues have emerged concerning whether the intended techniques have been properly implemented [8,9], and the problems associated with university students’ drinking persist.

Many interventions for university students’ drinking were designed to utilize the principles of Motivational Interviewing [10,11,12,13]. In the spirit of client-centered counseling [14], Motivational Interviewing is a nonconfrontational, supportive approach to counseling. When used with problem drinkers, Motivational Interviewing and its variants (Motivational Enhancement Therapy and the Drinker’s Check-Up) seek to support clients in choosing the drinking goal that is best for them. Specifically, these techniques (a) express empathy for the client’s feelings; (b) develop a discrepancy between where the client is now with respect to drinking and where the client would like to be; (c) roll with resistance, i.e., when the client resists the counselor’s suggestions, the counselor reflects on the client’s views and avoids argumentation; and (d) support clients’ self-efficacy by attempting to instill in them a feeling of self-worth and of being in control [15]. 

Evaluations of the effectiveness of MI and of MET for helping drinkers resolve their problematic use of alcohol have shown that both MI [16,17,18,19,20] and MET [21] are more effective than no treatment and are as effective as other treatments. Brief interventions with university students, which typically use Motivational Interviewing, most frequently report using change strategies that provide personal drinking summaries (including Blood Alcohol Concentration estimates), normative comparisons, consideration of the negative consequences of drinking, risk factors, and behavioral strategies [22], each of which was used in the interventions described in the present study.

Systematic Motivational Counseling (SMC) [23], on the other hand, is based on the motivational model of alcohol use [24,25,26,27,28]. SMC and its briefer variant—the Lifestyle Enrichment and Advancement Program [29,30]—start by assessing drinkers’ goals and concerns in various areas of their life (e.g., education, employment, finances, relationships) that might have an impact on their motivation to use alcohol. The technique seeks to identify and alter drinkers’ maladaptive motivational patterns that are sources of frustration or a determent to attaining emotional satisfaction without the use of alcohol. As Miller and Rollnick [11] described, “The structured goal attainment counseling procedures … nicely complement our more problem-focused discussions of motivational interviewing” (p. 188).

Systematic Motivational Counseling (SMC) begins by assessing participants’ motivational structure, using the Motivational Structure Questionnaire or one of its variants [28,31] to identify maladaptive motivational patterns that prevent drinkers from achieving emotional satisfaction, thereby propelling them to seek satisfaction by drinking alcohol. The faulty motivational patterns then become the focus of change during SMC. SMC has been shown to be effective in a variety of formats (with individuals, in groups, as self-help) and with a variety of samples of participants, including university students [29] and patients with diagnoses in addition to an alcohol use disorder [23,28].

The present study utilized the principles of both Motivational Interviewing and Systematic Motivational Counseling to address the problem of university students’ excessive use of alcohol. To do so, two computerized brief interventions were developed. One intervention—the Computerized Brief Intervention (CBI)—is based on the principles of Motivational Interviewing. The other intervention—the Computerized Brief Intervention-Enhanced (CBI-E)—is an enhanced version of the first intervention that includes (additional to the CBI intervention) components of Systematic Motivational Counseling (SMC). The addition of the SMC components for dealing with participants’ maladaptive motivation was important to include because maladaptive motivation is positively correlated with the amount of alcohol that university students consume [32,33]. As Miller and Rollnick [11] stated, the SMC goal-attainment procedures complement the alcohol-specific focus of MI.

The decision to use a researcher-delivered intervention employing a computer-assisted method was to ensure that (a) immediate graphic and printed feedback could be given, and (b) each participant would receive the same intervention method, albeit personalized for each person. 

The impact of each intervention on students’ drinking was compared with a control group. The purpose of the study was to evaluate the impact of two computerized motivational interventions compared to a control condition. We hypothesized that (a) compared to the control condition, reductions in drinking would be achieved with both CBI and CBI-E, and (b) the reductions achieved with CBI-E would be greater than those achieved with CBI. Because the overwhelming majority of prior studies evaluating alcohol use interventions for university students have not found gender differences (see [8,9,22,34], we did not expect to find gender differences and gender differences were not hypothesized. 

## 2. Materials and Methods

### 2.1. Participants

A power analysis indicated that with an expected attrition rate of 15% and an effect size of ƒ = 0.36, and to achieve a statistical power of 0.80 and *p* < 0.05 with three groups of participants, an initial sample of approximately 90 participants was needed. Miller et al. [22] reported medium-to-large effect sizes (η^2^ = 0.12 to 0.23) in their review of brief interventions with university students (i.e., proportionate to the Cohen’s *f* effect size of 0.36 selected for this study). Accordingly, 90 participants were recruited from students at Bangor University through an offer of either course credit (one hour of research credit for each hour that they participated) or a modest cash payment of GBP 10. The inclusion criteria were that men should drink more than 21 units of alcohol per week or 8 or more units on one occasion at least weekly, and women should drink more than 14 units per week or 6 or more units on one occasion at least weekly. Each UK unit of alcohol represents 10 mL or 8 g of pure alcohol. The participants were randomly assigned to one of the three groups: Computerized Brief Intervention (CBI), Computerized Brief Intervention-Enhanced (CBI-E), or the Control group. The final sample who completed the first session included 88 students, 54 (61.5%) of whom were female, and their mean age was 21.05 years (*SD* = 4.42). Seventy-five participants completed the follow-up session; 60% of them were female. In the CBI group (*n* = 27), 16 (60%) of the participants were female; in the CBI-E group (*n* = 22), 15 (68%) were female; in the Control group (*n* = 26), 14 (54%) were female. The three groups, therefore, were balanced with regard to gender. The mean age of participants who completed the follow-up session was 21.24 years (*SD* = 4.7). 

### 2.2. Baseline Assessment

At baseline, the participants were individually administered the following measures:

Quantity/Frequency Alcohol Consumption Questionnaire. The Quantity/Frequency (QF) Alcohol Consumption Questionnaire [35,36] is a simplified version of the Alcohol Timeline Followback (TLFB) [37] method for measuring alcohol consumption. The QF was used to record participants’ alcohol consumption during the 12 weeks prior to their inclusion in the study. Like the TLFB, the QF uses a calendar to assist the respondent in recalling the amounts of alcohol consumed in a given time period. Unlike the TLFB, the QF focuses on weekly drinking instead of daily drinking. The participant was asked to estimate (a) the usual amount of alcohol consumed during each week and the number of days this usual amount was consumed, and (b) the largest amount of alcohol consumed during each week and the number of days this largest amount was consumed. 

Drinker’s Inventory of Consequences: The Drinker’s Inventory of Consequences (DrInC–2R) [38] is a 50-item questionnaire that measures a variety of negative consequences of drinking that occurred during the prior three months. From respondents’ answers, both a total score and scores on five subscales depicting adverse consequences of drinking (physical, intrapersonal, social responsibility, interpersonal, impulse control) can be derived. The internal and test–retest reliability of the DrInC–2R has been confirmed [38]. 

Short Alcohol Dependency Data Questionnaire: The Short Alcohol Dependency Data Questionnaire (SADD) [39] is a 15-item questionnaire designed to estimate respondents’ level of alcohol dependency from their current drinking habits. Based on their total scores, respondents are classified as low, medium, or high in terms of dependency. Various studies [39,40] have confirmed the validity of the SADD.

Readiness to Change Questionnaire: The Readiness to Change Questionnaire (RTCQ) [41] is a 12-item questionnaire designed to assess respondents’ level of commitment to changing their drinking. Respondents are assigned to one of three stages of change—precontemplation, contemplation, or action—according to their highest total score. The total score is obtained from the respondent’s answers to the 12 items after reverse-scoring the answers to the precontemplation items [42]. Both the internal consistency and test–retest reliability of the RTCQ have been confirmed [43]. 

Short Tridimensional Personality Questionnaire: The Short Tridimensional Personality Questionnaire (Short–TPQ) is a 44-item questionnaire that measures three personality dimensions: Novelty Seeking (NS), Harm Avoidance (HA), and Reward Dependence (RD) [44]. The internal consistency and test–retest reliability of the three subscales were reported as good [44].

### 2.3. Computer Software

The computerized brief interventions used in the study were created on a Microsoft Excel 97–2004 Workbook (.xls) [45], which allowed various algorithms and images to be inserted into the program. 

### 2.4. Follow-Up Assessment

At follow-up, the participants were individually administered the Alcohol Timeline Followback [37], which retrospectively estimated their daily drinking during the prior three months with the use of a calendar to help respondents recall the amount of alcohol consumed each day. In addition, the participants were re-administered the DrInC–2R.

### 2.5. Procedure

At baseline, each participant was interviewed individually in a quiet experimental room at Bangor University. On arrival, the participant read and signed a consent form. In order to protect the anonymity and confidentiality of the participants, each person was assigned a number and data were kept securely in locked file cabinets in a locked office. The participants were randomly assigned to each group following a predetermined list according to the order in which they volunteered for the study. The participants’ alcohol consumption in all three groups was assessed with the QF, and they completed the other measures in the baseline battery before participants in the Control group were then dismissed and participants in the CBI group and the CBI-E group individually completed the computerized intervention. These participants received a printed summary of the results of the intervention.

Approximately 12 weeks (*M* = 85.1 days, *SD* = 7.8) later, all participants were interviewed about their drinking during the prior 12-weeks with the use of the Alcohol Timeline-Followback method, and they were also re-administered the DrInC–2R. Participants in the Control group were then offered the opportunity to complete the CBI intervention. Each participant was thanked, debriefed, and dismissed. The procedure that was implemented is shown in Figure 1. No data were missing from any of the questionnaires.

### 2.6. Interventions

#### 2.6.1. Computerized Brief Intervention

The participants were introduced to the CBI as a computer-assisted interview during which they would receive personalized and objective feedback about their drinking. The CBI aimed to motivate participants to change their current use of alcohol through the use of the following components: First, participants were given objective feedback about their drinking. Second, the discrepancy between participants’ current drinking and their drinking goal was highlighted. Third, participants were asked to consider the implications of not changing their current level of alcohol use. The CBI achieved these objectives in approximately 30 min through the use of 12 computer screens.

Screen 1: Welcome Page. Screen 2: Calculating Your Blood Alcohol Concentration. Screen 3: Your Weekly Alcohol Consumption (in comparison with other people’s drinking using gendered norms). Screen 4: Listing the Good Things About Drinking. Screen 5: Listing the Not-So-Good Things About Drinking. Screen 6: Weighing Up the Good Things and the Not-So-Good Things About Drinking. Screen 7: Future Consequences of Continuing to Drink in this Manner. Screen 8: Would You Like to Change Your Use of Alcohol? (yes, no, maybe). Screen 9: Your Drinking Goal (to cut down, stop, or stay the same). Screen 10: Strategies for Cutting Down or Stopping Drinking. Screen 11: Positive Action (to take today). Screen 12: Feedback Sheet (summarized the information from the previous screens).

#### 2.6.2. Enhanced Computerized Brief Intervention

Participants were introduced to the Enhanced Computerized Brief Intervention (CBI-E) as a computer-assisted task that was designed to give them both (a) personalized and objective feedback about their drinking, and (b) feedback about their concerns in other areas of their life that might have an impact on their drinking. The first 12 screens of the CBI-E were the same as those for the CBI. The number of additional screens was variable and depended on the number of concerns the participant named in the other areas; however, on average the additional screens took an additional 30 min to complete.

After completing the CBI, the participant was introduced to the Personal Concerns Inventory (PCI) [23,46], its rationale, and the steps necessary for completing it. The participants were then asked to select the PCI life areas (e.g., Home and Household; Employment and Finances; Relationships) in which they had concerns and to briefly describe each concern and their goal for resolving it. In subsequent screens, they rated each of their goal pursuits on ten rating scales (Commitment, Importance, How Likely to Succeed, Expected Joy, etc.). Doing so revealed the structure of the participant’s motivation (i.e., the degree to which it was adaptive or maladaptive), which was shown to the participant in graphical format. Thereafter, the interviewer provided feedback by calling attention to maladaptive patterns (e.g., pursuing unrealistic goals or goals that were unlikely to succeed or unlikely to be emotionally satisfying) and discussing with the participant ways to change faulty motivational patterns into adaptive ones. The suggestions were based on one of the 12 components of SMC [23]. Each suggestion aimed to help the participant find more adaptive ways to resolve important concerns that impacted his or her motivation to achieve emotional satisfaction by drinking alcohol or abstaining from drinking. For instance, it seemed best for some participants to disengage from goal pursuits that were unlikely to succeed or unlikely to be satisfying, even if successful. Some participants needed suggestions for finding pleasurable goals to pursue as an alternative to drinking alcohol. Other participants needed help in dividing their long-range goals into interim steps, whose achievement would bring a feeling of success. Still others seemed to benefit simply from receiving feedback about how their excessive drinking was interfering with their achievement of appropriate and realistic goals that they were already pursuing.

The interventions were conducted by the first author, who was an advanced Ph.D. student in clinical and health psychology. He was trained and supervised (a) in the delivery of the MI components by members of the counseling staff at Cyngor Alcohol Information Services, and (b) in the delivery of the SMC components by the second author, who is the co-developer of SMC.

### 2.7. Analysis

To analyze participants’ baseline characteristics, descriptive statistics, ANOVAs, Tukey post hoc tests, and *t*-tests were conducted. Similarly, to analyze changes in participants’ alcohol consumption following the interventions, ANOVAs and *t*-tests were used.

## 3. Results

### 3.1. Baseline Characteristics

At baseline, the male and female participants’ mean weekly alcohol consumption was 35.0 (*SD* = 22.7) units and 25.9 (*SD* = 18.1) units, respectively (*t*(86) = 2.097, *p* < 0.039). The mean number of binge episodes during the previous 12 weeks (defined as eight or more units on one occasion for males and six or more units for females) was 23.5 (*SD* = 12.7) for males and 22.61 (*SD* = 16.0) for females (*t*(86) < 1.0, *NS*). Table 1 displays the participants’ mean weekly alcohol consumption, along with the number of binge episodes and negative drinking consequences for each group.

The mean number of drinking-related negative consequences was 20.7 (*SD* = 11.5) for males and 23.2 (*SD* = 15.4) for females *(t*(86) < 1.0, *NS*). Had a hangover (at 95%) and felt bad about myself because of my drinking (at 66%) were the most frequently reported negative consequences. On the DrInC subscales, the participants scored highest on negative physical consequences, and next highest on adverse consequences related to social responsibility, impulse control, intrapersonal consequences, and interpersonal consequences.

To determine whether participants’ level of alcohol consumption was related to the number of alcohol-related problems they had experienced, they were allocated to one of three groups (low-risk drinkers, hazardous drinkers, or harmful drinkers), based on the UK’s public health guidelines [47]. A one-way analysis of variance indicated that the three groups differed significantly (*F*(2,87) = 8.240, *p* < 0.001). Tukey’s HSD post hoc tests showed that the low-risk drinkers had significantly lower DrInC total scores (*M* = 15.45, *SD* = 9.40) than both the hazardous drinkers (*M* = 21.46, *SD* = 11.65, *p* < 0.022) and the harmful drinkers (*M* = 33.81, *SD* = 18.82 *p* < 0.001), but the hazardous and harmful drinkers did not differ from each other. The participants’ scores on the SADD indicated that the majority of them (59.7%) were at the low level of alcohol dependence; 37.5% were at the medium level; and 3.4% were at the high level. 

The three groups (CBI, CBI-E, Control) were also compared on each of the other baseline measures, but there were no baseline differences. It is noteworthy, however, that females (*M* = 11.37, *SD* = 5.94) as a whole were significantly higher than males (*M* = 5.70, *SD* = 3.30) on the Harm Avoidance scale of the Short Tridimensional Personality Questionnaire (*t*(68) = 5.22, *p* < 0.001). The distribution of the three groups on the three stages of change (precontemplation, contemplation, action) from the Readiness to Change Questionnaire did not differ (χ^2^ (4, *n* = 88) = 2.34, *p* > 0.05). The majority of the participants were in either the precontemplation stage or the contemplation stage; only 22% of them were in the action stage.

### 3.2. Changes in Drinking at Follow-Up

The rate of attrition for the three groups was as follows: Control, *n* = 3 (10.3%); CBI *n* = 5; (15.6%); and CBI-E, *n* = 5 (17.2%). The mean weekly alcohol consumption of the participants who were retained in the study (*M* = 29.01, *SD* = 19.72) and those who dropped out (*M* = 30.62, *SD* = 23.06) did not differ (*t*(88) < 1.0, *NS*), and there was no apparent relationship between participants’ drop-out rates and their level of drinking at baseline. Reductions in mean weekly alcohol consumption from baseline to the three-month follow-up were examined for differences among the groups and, unexpectedly, differences between the male and female participants were detected. Consequently, for ease of interpretation, the three groups’ mean weekly alcohol consumption at baseline and during the follow-up period are shown separately for males and females in Figure 2 and Figure 3, respectively.

For males, after a steady reduction in drinking during the first three weeks after being included in the study, the weekly consumption of each of the groups stabilized. Individuals in the CBI-E group drank less alcohol than the other two groups, and the CBI group drank less than the Control group. For females, however, the post-intervention pattern of drinking was quite different. First, the female participants in all three groups considerably reduced their consumption just after they completed the baseline assessment. Second, the reductions were maintained throughout the 12 weeks of follow-up. Third, the differences in consumption among the three groups were less distinct than for the males.

Prior to analyzing the results statistically, the distributions of the variables were examined for deviations from normality, and any deviations were corrected with logarithmic or square root transformations. Thereafter, mean alcohol consumption during the 12 weeks prior to the interventions (baseline) was compared with mean consumption during the 12 weeks following the interventions (follow-up). A repeated-measures ANOVA that included groups (CBI, CBI-E, Control) and gender (females, males) as the between-participants factors and time of testing (baseline, follow-up) as the within-participants factor showed that the participants reduced their consumption from baseline (*M* = 29.0, *SD* = 19.7) to follow-up (*M* = 18.4, *SD* = 12.9, *F*(1,69) = 51.59, *p* < 0.001). There was also (a) a groups-by-time interaction (*F*(2,69) = 3.67, *p* < 0.04), indicating differential reductions among the three groups, and (b) a groups-by-time-by-gender interaction (*F*(2,69) = 3.26, *p* < 0.044).

To identify the source of the three-way interaction, repeated-measures ANOVAs were conducted separately for males and females. In each analysis, group was the between-participants factor, and time was the within-participants factor. For males, there was a main effect for time (*F*(1,27) = 23.92, *p* < 0.001), and a groups-by-time interaction (*F*(2,27) = 4.42, *p* < 0.022). The source of the two-way interaction was identified using paired-samples *t*-tests. For male participants in the CBI-E group, there was a significant reduction in alcohol consumption from baseline (*M* = 37.4, *SD* = 13.0) to follow-up (*M* = 14.6, *SD* = 7.4, *t*(6) = 3.69, *p* = 0.01). For male participants in the CBI group, there was also a significant reduction in alcohol consumption from baseline (*M* = 35.1, *SD* = 17.1) to follow-up (*M* = 22.7, *SD* = 14.4, *t*(10) = 3.07, *p* = 0.012). The male participants in the Control group, however, did not significantly reduce their consumption from baseline (*M* = 37.3, *SD* = 33.1) to follow-up (*M* = 29.3, *SD* = 18.3), *t*(11) = 0.96, *p* = 0.357. Additionally, the mean weekly consumption of the males who received the CBI-E intervention was below the UK Department of Health’s recommended limit of 21 units per week for each of the 12 weeks following the intervention. The males in the CBI group were below this limit during just four of these weeks. The males in the Control group, on the other hand, exceeded this limit during each of the 12 weeks.

Females also reduced their consumption from baseline (*M* = 23.9, *SD* = 14.9) to follow-up (*M* = 15.1, *SD* = 9.5, *F*(1,42) = 29.081, *p* < 0.001). However, the groups-by-time interaction (*F*(2,42) = 1.36, *p* = 0.27) was not significant, indicating that the three groups did not differ in the amount they reduced from baseline (CBI-E group: *M* = 23.0, *SD* = 9.9; CBI group: *M* = 25.3, *SD* = 20.9; Control group: *M* = 23.3, *SD* = 12.1) to follow-up (CBI-E group: *M* = 17.6, *SD* = 12.3; CBI group: *M* = 12.1, *SD* = 7.2; Control group *M* = 15.7, *SD* = 8.2).

### 3.3. Reductions in Binge Drinking

A repeated-measures ANOVA was also used to evaluate the changes in the number of binge-drinking episodes from baseline to follow-up. The participants as a whole reduced the number of binge episodes from baseline (*M* = 22.2, *SD* = 13.6) to follow-up (*M* = 14.7, *SD* = 11.1, *F*(1,69) = 30.46, *p* < 0.001). Because the groups-by-time interaction closely approached significance (*F*(2,69) = 3.07, *p* < 0.053), paired-samples *t*-tests were run to identify the source of the interaction. The participants in the CBI-E group reduced the number of binge episodes from baseline (*M* = 22.59, *SD* = 10.17) to follow-up (*M* = 13.95, *SD* = 11.81, *t*(21) = 3.86, *p* = 0.001). Likewise, the participants in the CBI group reduced the number of binge episodes from baseline (*M* = 24.04, *SD* = 17.86) to follow-up (*M* = 13.30, *SD* = 10.16, *t*(26) = 4.17, *p* < 0.001). The participants in the Control group, however, did not significantly reduce the number of binge episodes from baseline (*M* = 19.88, *SD* = 11.07) to follow-up (*M* = 16.85, *SD* = 11.40, *t*(25) = 1.44, *p* = 0.16). Neither the time-by-gender interaction (*p* = 0.688) nor the three-way interaction (*p* = 0.139) was significant.

Changes in the number of negative consequences of drinking from baseline to follow-up were also examined for differences between groups and between genders. A repeated-measures ANOVA yielded a significant main effect for time, indicating that the number of negative consequences decreased from baseline (*M* = 22.1, *SD* = 14.1) to follow-up (*M* = 17.6, *SD* = 11.2, *F*(1,67) = 11.08, *p* < 0.001). There was also a gender-by-time interaction (*F*(1,67) = 4.25, *p* < 0.043). Males did not change (*M* = 20.5 (*SD* = 12.1) between baseline and (*M* = 19.7, *SD* = 11.7) at follow-up (*t*(28) = < 1.0, *NS*). By contrast, females decreased from *M* = 23.1 (*SD* = 15.4) at baseline to *M* = 16.3 (*SD* = 10.8) at follow-up (*t*(43) = 3.99, *p* < 0.001).

## 4. Discussion

Two brief interventions aimed at reducing university students’ excessive use of alcohol were evaluated. The Computerized Brief Intervention (CBI) utilized the principles of Motivational Interviewing to directly address students’ excessive use of alcohol and motivate them to change their drinking. The Enhanced Computerized Brief Intervention (CBI-E) additionally addressed students’ concerns in other life areas that affected their motivation to drink alcohol. Consistent with Value X Expectancy theory [27,48,49,50], the aim was to help students form goals that both held value for them and for which they had a realistic expectancy of successful goal attainment. We hypothesized that both CBI and CBI-E would induce a reduction in consumption and that CBI-E would bring about a greater reduction than CBI.

Unexpectedly, distinct differences between the male and female participants’ responsiveness to the two interventions were revealed. The results obtained with male participants supported the research hypotheses. Unlike male participants in the Control group, those in both the CBI-E group and the CBI group reduced their consumption, and the reduction in the CBI-E group exceeded that of the CBI group. The post-intervention pattern of drinking of the female participants was, however, quite different. The females in all three groups (CBI-E, CBI, Control) reduced their consumption, and the reduction was relatively stable throughout the post-intervention period.

The gender difference in responsiveness to the two interventions can be accounted for in the following way: at the baseline assessment, the participants in each of the three groups answered detailed questions about their alcohol consumption during the preceding three months. The assessment is likely to have made participants acutely aware of how much alcohol they were drinking; the negative consequences they had experienced because of their excessive drinking; and the need to cut down. The explanation for why only the female participants reacted to the assessment by reducing their drinking appears to lie in a personality difference between the male and female participants. Consistent with the findings of other studies [51,52], the female participants in this study scored higher on harm-avoidance than the male participants. Harm-avoidant individuals use protective behavioral strategies to avoid engaging in harmful behaviors when they become aware of them. Therefore, for the female participants, the assessment itself appeared to serve as a brief intervention that induced them to reduce their alcohol consumption.

In previous studies, a gender difference in university students’ responsiveness to a brief intervention for excessive drinking has often not been found [8,9,22,34,53,54]. Nevertheless, within other samples the gender difference that we found is not without precedent. Specifically, the World Health Organization’s study of brief interventions [55] reported that females in both the control group and the intervention group reduced their alcohol consumption. Males, on the other hand, reduced their alcohol consumption only if they received a brief intervention. Similarly, Chang [56] reviewed prior studies of brief interventions with females and concluded that brief interventions were not consistently more helpful for females than a control condition. Other studies [57,58] have found a gender difference in changes in drinking, although the changes were not a function of the intervention received.

The results of this initial evaluation of CBI and CBI-E are promising for two reasons. First, it is noteworthy that for the female participants, the comprehensive assessment of their drinking habits was sufficient to cause a stable reduction in consumption, and this was apparently due to the females’ stronger use of protective behavioral strategies. Second, the study also demonstrated that when used with male university students, both CBI and CBI-E were effective, but CBI-E was more effective than CBI. The improved outcome with CBI-E was theoretically expected because the enhanced intervention addressed both participants’ problematic use of alcohol and their maladaptive motivational patterns that undergirded their motivation to drink alcohol.

### Limitations

Despite these encouraging results, the limitations of the current study should be addressed in future evaluations of the two motivational interventions. First, although a power analysis indicated that the sample size was sufficient for testing the main hypotheses, the unexpected need to divide the sample into subgroups in order to evaluate gender differences resulted in relatively small subgroups. Second, in future studies, it would be advisable to use additional controls, particularly a sham intervention to determine the extent to which the better outcome achieved with the enhanced intervention was due to the additional attention that the CBI-E group received. Third, instead of asking participants to retrospectively recall their drinking during each week of the preceding three months, a more accurate record of their drinking might be achieved by asking participants to keep a prospective daily diary of their drinking. Fourth, this study was conducted in the United Kingdom, and it is not known how generalizable the findings are to other countries and other cultures with different drinking practices. Finally, it would be worthwhile in future to explore different combinations of the brief computerized interventions. For instance, how would the enhanced components of CBI-E alone fare as a brief intervention for university students when used after a comprehensive assessment of participants’ drinking behavior? Also, would the enhanced components used alone be more effective for certain types of students (e.g., males and females who use protective behavioral strategies) than others?

## Figures and Tables

**Figure 1 behavsci-14-00381-f001:**
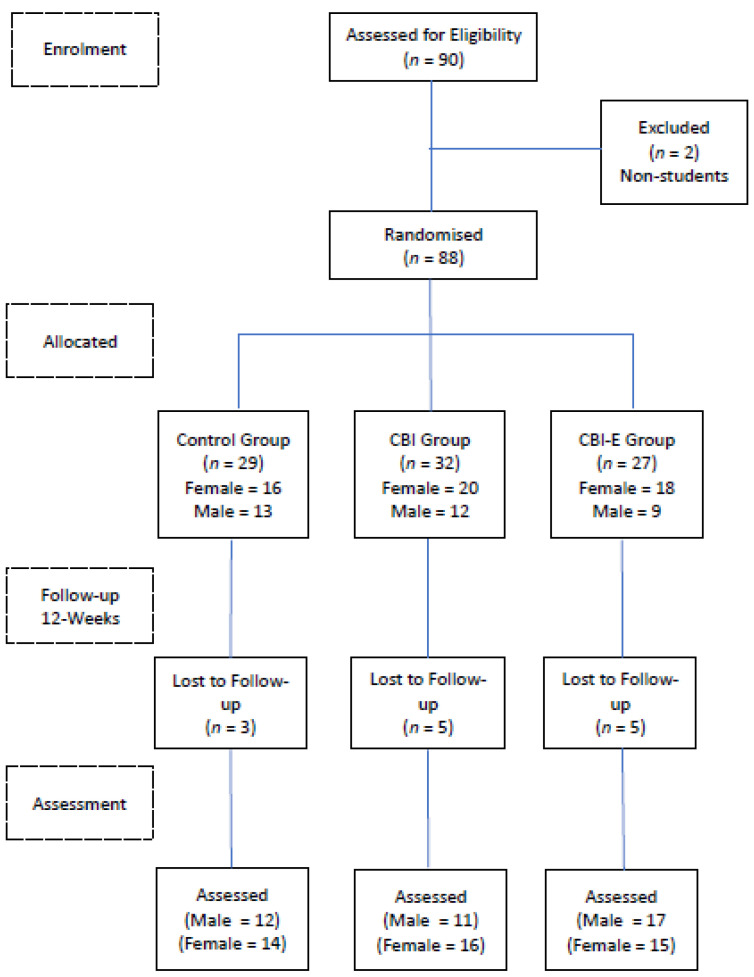
Flow diagram of implemented study procedure.

**Figure 2 behavsci-14-00381-f002:**
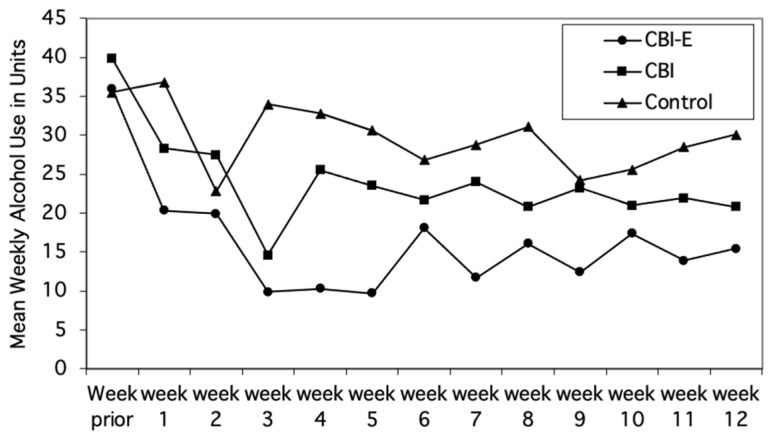
Mean weekly alcohol consumption of males in the CBI-E, CBI, and Control group during the week prior to being included in the study and during each of the 12 weeks following the intervention.

**Figure 3 behavsci-14-00381-f003:**
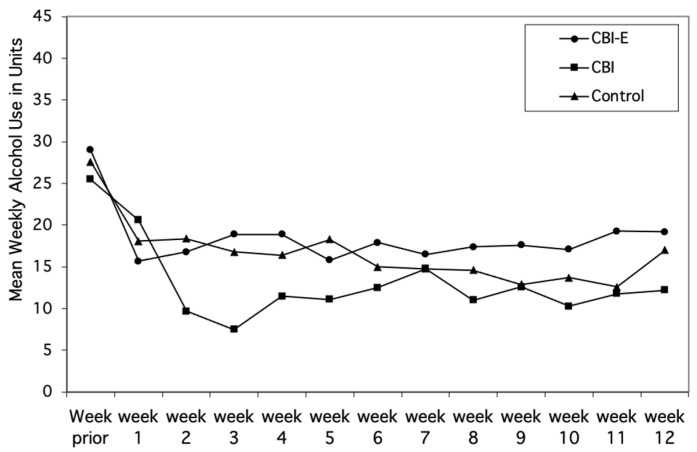
Mean weekly alcohol consumption of females in the CBI-E, CBI, and Control group during the week prior to being included in the study and during each of the 12 weeks following the intervention.

**Table 1 behavsci-14-00381-t001:** Means and standard deviations of weekly alcohol consumption, number of binges, and drink-related problems at baseline (t1) and follow-up (t2) for males and females in the control, CBI, and CBI-E groups.

	Control				CBI				CBI-E			
	Male (*n* = 12)		Female (*n* = 14)		Male (*n* = 11)		Female (*n* = 16)		Male (*n* = 7)		Female (*n* = 15)	
Variable	*M*	*SD*	*M*	*SD*	*M*	*SD*	*M*	*SD*	*M*	*SD*	*M*	*SD*
Weekly units, t1	37.3	33.1	23.3	12.1	35.1	17.1	25.3	20.9	37.4	13.0	23.0	9.9
Weekly units, t2	29.3	18.3	15.7	8.2	22.7	14.4	12.1	7.2	14.6	7.4	17.6	12.6
Binge ^†^ total, t1	20.1	13.5	19.7	9.0	25.0	12.8	23.4	21.0	27.7	12.8	20.2	8.1
Binge ^†^ total, t2	18.9	12.7	15.1	10.3	17.5	10.7	10.4	9.0	9.8	6.5	16.0	13.3
DrInC total, t1	20.8	14.9	25.3	14.5	18.4	7.8	22.9	19.0	23.3	13.5	21.4	12.4
DrInC total, t2	20.3	13.3	15.4	8.2	19.4	11.5	14.6	11.6	19.1	10.6	18.8	12.1

Note: ^†^ Binge criteria were that males consumed eight units or more on one occasion, and females, six units or more on one occasion.

## Data Availability

Data supporting the results can be obtained by contacting L.M.H. at lee.hogan@bangor.ac.uk.

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
