# Peer review of "Evaluating Two Brief Motivational Interventions for Excessive-Drinking University Students"

_behavsci, 2024, doi:10.3390/bs14050381_

Round 1
Reviewer 1 Report
Comments and Suggestions for Authors
See attachment

There is an excessive use of additional words that do not need to be included. The writing could be tapered, providing more concise sentences and clearer writing. I noticed many misspelled words throughout the manuscript. The writing goes back and forth between past and present tense. The authors use a lot of first person, which is generally not encouraged. Authors should consider rewriting in third person to make the manuscript more robust. For some reason, particularly in the latter half of the manuscript, the authors pose a lot of questions to the readers. This detracts from the robust nature of the study. Authors would benefit from refraining from asking questions and writing their work in the form of statements.
Author Response
There is an excessive use of additional words that do not need to be included. The writing could be tapered, providing more concise sentences and clearer writing. I noticed many misspelled words throughout the manuscript. The writing goes back and forth between past and present tense. The authors use a lot of first person, which is generally not encouraged. Authors should consider rewriting in third person to make the manuscript more robust. For some reason, particularly in the latter half of the manuscript, the authors pose a lot of questions to the readers. This detracts from the robust nature of the study. Authors would benefit from refraining from asking questions and writing their work in the form of statements.
We have carefully reread the manuscript and eliminated any unnecessary words.
We did not find misspelled words. Perhaps the reviewer is a speaker of British English and is reacting to our use of American spelling with which he or she might not be familiar. For instance, Reviewer 2 suggested that when we wrote insure we meant ensure instead. Insure is American; ensure is British; nevertheless, in the manuscript we changed insureto ensure. Similarly, counseling is American; counselling is British. Systematic Motivational Counseling, which is cited in the manuscript, is a proper noun, which must retain the American spelling that its developers used.
Our use of verb tenses is entirely consistent with the rules of English grammar. For instance, we used past tense when we referred to events that happened in the past; we used present tense when we referred to ongoing events in the present.
Some years ago, the use of first-person pronouns was discouraged, but according to the 7th Edition of the Publication Manual of the American Psychological Association, that is no longer the case, e.g., “APA format suggests writing in the first person. If you are co-authoring a paper, use the pronoun we. Do NOT refer to yourself or your co-authors in the third person as the author(s) or the researcher(s).”
Finally, we have changed the two rhetorical questions that we included in the Discussion to declarative sentences.
A definition of binge drinking and excessive drinking should be provided to reinforce the hazardous nature. Line 32
In the manuscript, we have added a definition of both excessive drinking and binge drinking. Lines 32 - 34
In this line the authors switch to the term “heavy-drinking.” In order to remain consistent, I encourage the authors to use the same terminology throughout the paper. For example, use “excessive drinking.” Line 35
We have changed heavy-drinking students to excessive-drinking students. Line 37
Motivational Enhancement Therapy and the Drinker’s Check-Up should have reference if it is a published instrument. Line 46
The Production Editor cautioned us about adding references to the manuscript because it has already been typeset. Accordingly, we have opted not to add references for Motivational Enhancement Therapy and the Drinker’s Check-Up. Readers who are familiar with motivational interviewing should be familiar with these two variants, or they can easily locate the references themselves.
It is not recommended to use such a large quote within a manuscript. I suggest you revise this reference and paraphrase with proper citation. Lines 68-76
We have reduced the long quotation to one sentence. Lines 74-76
I believe the word you are looking for in this sentence is “ensure” and not “insure”. Line 98
We intentionally used insure as the American spelling; nevertheless, we have changed it to ensure. Line 98
There is no purpose statement in the introduction. I recommend writing a purpose statement to make the reason for this research clearer to the reader.
An additional sentence has been added to the final paragraph of the introduction to outline the purpose of the study. Lines 123-124
What software was used? This should be included and correctly cited within text and in the reference page.
A new section in Materials and Methods 2.4 outlines the software that was used. Lines 201-204
Why were the alcohol consumption inclusion criteria needed? Lines 117-120
The inclusion criteria were necessary to ensure that participants were, in fact, excessive drinkers who could potentially benefit from having a brief intervention for excessive drinking.
Based on the first sentence (lines 110-112) the authors needed 90 participants in order to ensure efficacious results. However, the authors only obtained 88 participants. I am not sure how these results could contribute to literature if the authors did not collect the minimum number of participants they indicated were needed. I suggest revising lines 110-112. It seems like 90 participants is somewhat arbitrary because in a study of this scope with an administered intervention, the 88 should suffice. Line 123
Reviewer 2 is entirely correct that the N of 90 suggested by the power analysis was an approximation. We now indicate that approximately 90 participants were needed. Line 134
Standard deviations should be abbreviated in all capital letters such that it reflects SD and not sd as it is currently written in the manuscript.
We have changed all lowercase sds to uppercase SDs.
This is a longitudinal study and should be specified.
By definition, “Longitudinal studies employ continuous or repeated measures to follow individuals over prolonged periods of time—often years or decades”. Our study included a three-month follow-up; hence, it does not quality as a longitudinal study.
How was autonomy of personally private and sensitive information maintained?
Customary procedures were followed, i.e., participant numbers rather than names were used, and data were kept securely in locked file cabinets in a locked office. This has now been added to the manuscript. Lines 153-157
Are the average consumption results consistent with existing literature? Lines 255-259
Participants’ mean consumption at baseline is consistent with other studies of university students’ drinking that we have conducted at Bangor University.
This section seems more like a limitations section. Furthermore, the authors pose questions in this section which is an interesting and nontraditional approach. However, it does not lend itself toward quality. I recommend reorganizing to fit into the limitations. Lines 410-422
To the final paragraph in the Discussion, which pertains to limitations, we have added a subheading, viz., 4.1 Limitations. The two rhetorical questions that we asked earlier in the Discussion do not pertain to limitations. We have, however, changed the two interrogative sentences to declarative sentences. Line 612
The authors draw a causal inference by using the term “apparently.” Authors should explain in further detail, why females have stronger protective behavioral strategies? Is there existing evidence that shows this to be true? Line 437
We explained why females have stronger protective behavioral strategies than males. First, the baseline results of the current study indicated that the female participants were higher on harm-avoidance, and harm-avoidant individuals use protective behavioral strategies to avoid engaging in harmful behaviors when they become aware of them. Second, we had already cited two additional studies (Braitman et al., 2017; Gierski et al., 2017) confirming that females are higher than males on harm-avoidance.
What is a “sham intervention?” And why would it benefit future studies? Line 449
In clinical research, a sham intervention is commonly used when the effects of a new intervention are being investigated. Although inactive, it is designed to mimic as closely as possible an active intervention being studied in a clinical trial. Its purpose is to evaluate the placebo effects of the new intervention. In line with the Production Editor’s caution about adding references to the typeset manuscript (described above), we have elected not to add a reference for sham interventions.
What are some examples of varying cultural contexts that could have led to the results in this study? Lines 454-455
As we indicate, it is not known how generalizable the current findings are to other countries and other cultures with different drinking practices. For instance, the interventions that we evaluated might be either more or less effective in cultures with more liberal or more restrictive drinking practices that those in the United Kingdom.
Reviewer 2 Report
Comments and Suggestions for Authors
Comments to Authors
The present study, “Evaluating two brief motivational interventions for excessive drinking university students,” evaluated two brief computerized motivational interventions for excessive drinking university students. The results showed that there appeared to be sexual differences in excessive drinking. Male alcohol use was ordered CBIE < CBI < Controls. Females showed nonsignificant differences in computerized motivational intervention for excessive drinking. The study is interesting. However, it has some statistical errors as follows.
1. The statistical analysis should be written in the Method section.
2. Table 1 should be analyzed by 2 x 3 x 6 mixed three-way (sex vs. group vs. weeks) ANOVA. Later, two-way ANOVA was conducted. If needed, one-way ANOVA was performed. By the way, table 1’s data were inference statistics but not description statistics. Thus, the authors did not use SD for variance. It should be shown as SE (standard errors).
3. Figure 2 needs 3 x 13 mixed two-way ANOVA for weekly alcohol use in units. Later, one-way ANOVA was performed. When appropriate, the post hoc with Tukey should be tested.
4. Figure 3 also had the same statistical problems as Figure 2. By the way, Figure 3 should be re-drawn as Figure 2. The x-axis of Figure 3 needs to be revised.
In summary, because statistical problems may influence the Results and Discussion sections, it was considered to be majorly revised for the Results and Discussion sections. The current status of the present study is not good enough for publication. The study should be considered for major revision.
Comments on the Quality of English Language
Dear authors,
Greetings for the day.
The English quality needs to be slightly revised.
Thank you.
Andrew
Author Response
Dear Reviewer, Please find our responses to your comments below. We are very grateful for your thought-provoking comments. We hope that you find that the quality of the manuscript is much improved with our efforts to address the quality of language.
- The statistical analysis should be written in the Method section.
We have added a subsection (2.8 Analysis) at the end of the Methods section to describe the statistical analyses that were conducted.
- Table 1 should be analyzed by 2 x 3 x 6 mixed three-way (sex vs. group vs. weeks) ANOVA. Later, two-way ANOVA was conducted. If needed, one-way ANOVA was performed. By the way, table 1’s data were inference statistics but not description statistics. Thus, the authors did not use SD for variance. It should be shown as SE (standard errors).
We chose not to analyze the data presented in Table 1 as a 2 x 3 x 6 (sex x groups x weeks) ANOVA for the following reasons: First, weeks is not a variable in the data in Table 1. Second, the only “6” in the table is indicated by the 6 subgroups (2 sexes in each of 3 groups = 6). However, both sex and groups are already included as between-participant factors in the ANOVA that we report in the manuscript.
- Figure 2 needs 3 x 13 mixed two-way ANOVA for weekly alcohol use in units. Later, one-way ANOVA was performed. When appropriate, the post hoc with Tukey should be tested.
The main goals of the analysis of the data shown in Figure 2 were (a) to determine whether there were significant reductions in participants’ consumption from before to after the interventions were introduced, and (b) whether there were differential reductions in the three intervention groups. In a 3 x 13 analysis, participants’ baseline drinking during the 12 weeks prior to the intervention would be represented by only one data point, whereas post-intervention drinking would be indicated by 12 data points. This overrepresentation of the post-intervention drinking relative to baseline drinking would mask the reductions that occurred, while focusing on the week-to-week variations in post-intervention drinking. Although the week-by-week fluctuations in drinking are interesting to see graphically, our main goal was to determine whether there was an overall reduction from before to after the interventions. This goal is realized by the analysis reported in the manuscript.
- Figure 3 also had the same statistical problems as Figure 2. By the way, Figure 3 should be re-drawn as Figure 2. The x-axis of Figure 3 needs to be revised.
The same rationale that we present for the analysis conducted for the data illustrated in Figure 2 applies to the analysis conducted for the data illustrated in Figure 3. The reviewer is entirely right that the labels for the x-axis in Figure 2 and Figure 3 should be indicated in the same way, and we have made this change.
Round 2
Reviewer 1 Report
Comments and Suggestions for Authors
I have no additional comments or suggestions.
Author Response
Many thanks for your initial suggestions on our manuscript.
Reviewer 2 Report
Comments and Suggestions for Authors
Author's Notes
Dear Reviewer, Please find our responses to your comments below. We are very grateful for your thought-provoking comments. We hope that you find that the quality of the manuscript is much improved with our efforts to address the quality of language.
- The statistical analysis should be written in the Method section.
We have added a subsection (2.8 Analysis) at the end of the Methods section to describe the statistical analyses that were conducted.
My comment: the response is ok.
- Table 1 should be analyzed by 2 x 3 x 6 mixed three-way (sex vs. group vs. weeks) ANOVA. Later, two-way ANOVA was conducted. If needed, one-way ANOVA was performed. By the way, table 1’s data were inference statistics but not description statistics. Thus, the authors did not use SD for variance. It should be shown as SE (standard errors).
We chose not to analyze the data presented in Table 1 as a 2 x 3 x 6 (sex x groups x weeks) ANOVA for the following reasons: First, weeks is not a variable in the data in Table 1. Second, the only “6” in the table is indicated by the 6 subgroups (2 sexes in each of 3 groups = 6). However, both sex and groups are already included as between-participant factors in the ANOVA that we report in the manuscript.
My comments: This point was not fully responded to my comments.
- Figure 2 needs 3 x 13 mixed two-way ANOVA for weekly alcohol use in units. Later, one-way ANOVA was performed. When appropriate, the post hoc with Tukey should be tested.
The main goals of the analysis of the data shown in Figure 2 were (a) to determine whether there were significant reductions in participants’ consumption from before to after the interventions were introduced, and (b) whether there were differential reductions in the three intervention groups. In a 3 x 13 analysis, participants’ baseline drinking during the 12 weeks prior to the intervention would be represented by only one data point, whereas post-intervention drinking would be indicated by 12 data points. This overrepresentation of the post-intervention drinking relative to baseline drinking would mask the reductions that occurred, while focusing on the week-to-week variations in post-intervention drinking. Although the week-by-week fluctuations in drinking are interesting to see graphically, our main goal was to determine whether there was an overall reduction from before to after the interventions. This goal is realized by the analysis reported in the manuscript.
My comments: OK.
- Figure 3 also had the same statistical problems as Figure 2. By the way, Figure 3 should be re-drawn as Figure 2. The x-axis of Figure 3 needs to be revised.
The same rationale that we present for the analysis conducted for the data illustrated in Figure 2 applies to the analysis conducted for the data illustrated in Figure 3. The reviewer is entirely right that the labels for the x-axis in Figure 2 and Figure 3 should be indicated in the same way, and we have made this change.
My comments: OK.
Author Response
Dear Reviewer, you have indicated that we had not fully responded to your comments about our Table 1. We see now that we inadvertently omitted our response to your suggestion that the data presented in Table 1 are inferential statistics. Means and standard deviations, like those presented in Table 1, are descriptive statistics. By contrast, inferential statistics use statistical tests (e.g., t-tests, analysis of variance) to assess the probability with which data drawn from samples can be generalized to the population from which the samples were drawn. The means and standard deviations presented in Table 1 are descriptive statistics. Apologies for missing this aspect from our response.